# Water mediated redox-neutral cleavage of arylalkenes via photoredox catalysis

Ke Liao[1,4], Yuqi Fang[1,2,4], Lei Sheng[3], Jiean Chen [1] ✉ & Yong Huang [2] ✉

Cleavage of carbon-carbon bonds remains a challenging task in organic synthesis. Traditional methods for splitting $C_{sp2}=C_{sp2}$ bonds into two halves typically involve non-redox (metathesis) or oxidative (ozonolysis) mechanisms, limiting their synthetic potential. Disproportionative deconstruction of alkenes, which yields one reduced and one oxidized fragment, remains an unexplored area. In this study, we introduce a redox-neutral approach for deleting a $C_{sp2}$ carbon unit from substituted arylalkenes, resulting in the formation of an arene (reduction) and a carbonyl product (oxidation). This transformation is believed to proceed through a mechanistic sequence involving visible-light-promoted anti-Markovnikov hydration, followed by photoredox cleavage of $C_{sp3}$-$C_{sp3}$ bond in the alcohol intermediate. A crucial consideration in this design is addressing the compatibility between the highly reactive oxy radical species in the latter step and the required hydrogen-atom-transfer (HAT) reagent for both steps. We found that ethyl thioglycolate serves as the optimal hydrogen-atom shuttle, offering remarkable chemoselectivity among multiple potential HAT events in this transformation. By using $D_2O$, we successfully prepared dideuteromethylated (-$CD_2H$) arenes with good heavy atom enrichment. This work presents a redox-neutral alternative for alkene deconstruction, with considerable potential in late-stage modification of complex molecules.

Carbon-carbon double bonds are among the most prevalent functional groups found in both natural and man-made chemical feedstocks[1,2] Consequently, the chemical derivatization of alkenes has become a heavily investigated area in organic synthesis[3–8] The cleavage of $C_{sp2}=C_{sp2}$ bonds offers a powerful approach for transforming olefins into other functional groups[9] Transition-metal-catalyzed olefin metathesis has found extensive applications in the synthesis of natural products and biologically relevant functional molecules[10,11] Ozonolysis represents another potent method capable of effectively cleaving a double bond into two carbonyl fragments[12–14]. Other oxidative methods, such as osmium tetroxide and potassium permanganate, can achieve similar outcomes[15–17]. However, these traditional oxidative methods often employ reagents of safety concerns, prompting recent

efforts to identify milder oxidants for alkene cleavage[18–22]. Notably, visible light photocatalysis has emerged as a potential solution to replace ozone and other strong oxidants. Simonetti, Leonori and Parasram used purple-light irradiation and nitroarenes to generate 'N-doped' ozonides, which can be hydrolyzed under mild conditions to yield the corresponding carbonyl products[23,24] Despite these advances, current oxidative methods are limited to converting alkenes into two carbonyl halves (Fig. 1a). Developing a redox-neutral, disproportionative protocol capable of converting alkenes into both a reduction and an oxidation fragment would be highly desirable and represents a significant challenge in the field.

In this article, we introduce a mild, redox-neutral method to cleave arylalkenes, yielding an arene (reduction fragment) and a

[1]Pingshan Translational Medicine Center, Shenzhen Bay Laboratory, Shenzhen 518118, China. [2]Department of Chemistry, The Hong Kong University of Science and Technology, Clear Water Bay, Kowloon, Hong Kong SAR, China. [3]College of Pharmacy, Shenzhen Technology University, Shenzhen 518118, China. [4]These authors contributed equally: Ke Liao, Yuqi Fang. ✉e-mail: chenja@szbl.ac.cn; yonghuang@ust.hk

**Fig. 1 | Cleavage of alkenes. a** Traditional methods of alkene cleavage typically involve oxidative processes, which allow for the formation of two carbonyl fragments. **b** We herein report a redox-neutral approach to cleave $C_{sp2}=C_{sp2}$ bonds. This process involves a sequence of visible-light-promoted anti-Markovnikov hydration, followed by the redox cleavage of the $C_{sp3}$-$C_{sp3}$ bond. PC photocatalyst, PCET photo-coupled electron transfer, BHAT back hydrogen-atom transfer.

carbonyl product (oxidation fragment, Fig. 1b) in a single step. Our strategy combines a visible-light-promoted, anti-Markovnikov hydration step, pioneered by Nicewicz and co-workers[25,26], with an oxygen radical-mediated β-scission, inspired by our recent progress in C-C bond cleavage reactions[27,28]. However, the inherent incompatibility of these two chemical events poses a significant challenge. The anti-Markovnikov water addition to arylalkenes necessitates a hydrogen-atom-transfer (HAT) reagent to intercept a benzyl radical intermediate, completing the catalytic cycle. Conversely, the following step requires an oxygen radical to perform the β-scission. Due to the high bond dissociation energy of O-H bonds, the oxygen radical is expected to undergo rapid BHAT with the HAT reagent, neutralizing the intended functions of both[29,30]. In our previous studies, we observed a significant decrease in the efficiency of β-scission in the presence of a HAT reagent. The key to successfully executing this challenging sequence lies in identifying suitable conditions that allow the oxygen radical to have a sufficient half-life for the bond cleavage before reverse HAT occurs. Following the β-scission, a benzyl radical and an aldehyde/ketone are produced. The HAT reagent is again required to reduce this radical species into an arene. Notably, no stoichiometric oxidant is needed in this transformation. By replacing water with $D_2O$, we can access $CD_2H$-substituted arene products. In contrast to traditional alkene cleavage methods, which yield two identical functional groups, our complementary approach simultaneously reduces one fragment and oxidizes another, generating two distinct classes of products.

## Results

Upon conducting a rapid screening of photocatalysts (PCs), we identified a variant of the Fukuzumi-Nicewicz acridinium salt **PC1** as an effective PC for converting arylalkenes into the corresponding arene and ketone products (see Supplementary Information for more details)[31,32]. The reaction was optimally performed under irradiation from a 440 nm blue LED, and slightly elevated temperatures (35 °C)

were found to be beneficial. Ethyl thioglycolate ($EtO_2CCH_2SH$) was identified as the best HAT reagent for this transformation. Low yields were observed for other thiols (see Supplementary Information for a detailed comparison). Tetra-, tri-, and 1,2-disubstituted alkenes underwent smooth C=C cleavage, while terminal olefins led to complex mixtures[33]. Control experiments confirmed that water and the HAT reagent are essential for the reaction. No reaction took place when the mixture was stirred in the dark or in the absence of **PC1**, indicating the photoredox nature of the process.

Utilizing the optimized conditions, we explored the scope of arylalkenes for the C=C bond cleavage. In Fig. 2, we outline the scope of the reduced carbon (arene product). Tetra-substituted alkenes bearing various functional groups underwent the desired bond cleavage efficiently, yielding substituted ethylbenzene products (Fig. 2, products **1–5**). Benzothiazole and benzoxazole analogs were also found to be compatible with the reaction conditions. Removing the R group resulted in the formation of substituted toluene products with comparable yields (products **8–15**). Tolerance to various functional groups was examined. The reaction failed to proceed when a primary or a secondary alcohol group was present. In contrast, tertiary alcohols reacted smoothly to yield the desired arene products in high yield (product **16**). Alkynes were also compatible despite their high reactivity towards radicals. Ketones and esters did not interfere with the intended bond cleavage. Basic amines were to be incompatible due to their low oxidation potentials. In contrast, sulfonamide was tolerated. We subsequently tested substrates containing both a styrenyl and an aliphatic alkene. The results indicated that good chemoselectivity can be achieved for styrene over aliphatic alkene (product **22**), although the isolated yield was moderate. We also observed selectivity towards styrenyl over electron-deficient olefins (product **21**). However, no selectivity was observed between two electronically biased styrenes, which resulted in messy mixtures. The R substituent can be extended well beyond methyl (products **23–28**), with primary, secondary, and

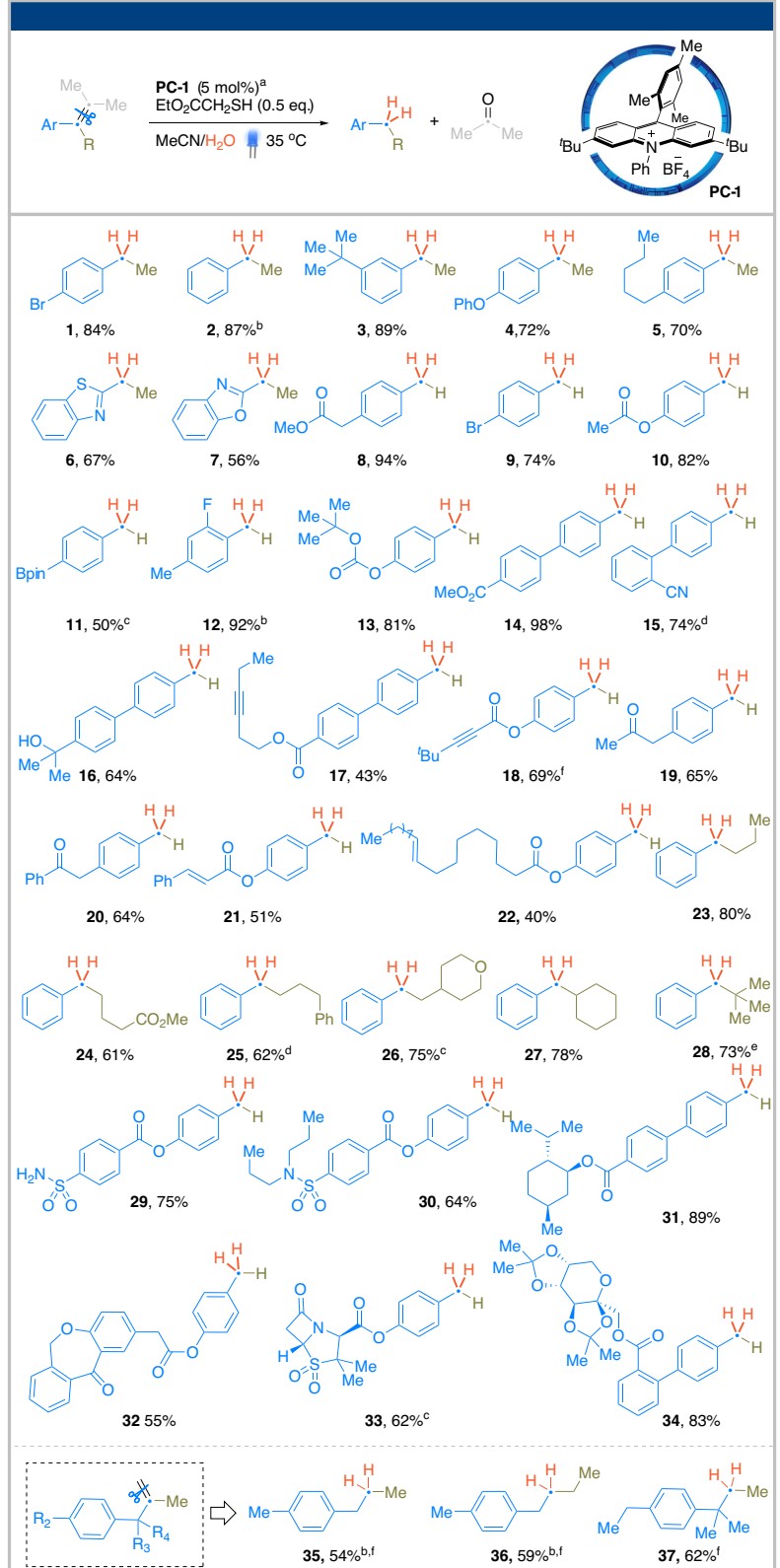

**Fig. 2 | Scope of redox-neutral arylalkene cleavage: the reduced fragment.** [a]A two-dram glass vial containing a mixture of **PC-1** (0.01 mmol, 5 mol%), arylalkene (0.2 mmol), EtO₂CCH₂SH (0.1 mmol) in MeCN/water (1.0 mL/0.1 mL) was placed in a 35 °C water bath and irradiated using two Kessil LED lightbulbs (440 nm, 40 W) under vigorous stir. The standard reaction time was 24 h. [b]The yield was determined by GC due to the low boiling point of these products. [c]The reaction time was 48 h. [d]The reaction time was 72 h. [e](4,4-Dimethylpent-2-en-3-yl)benzene was used as the substrate. [f]Pentafluorothiophenol was used as HAT reagent.

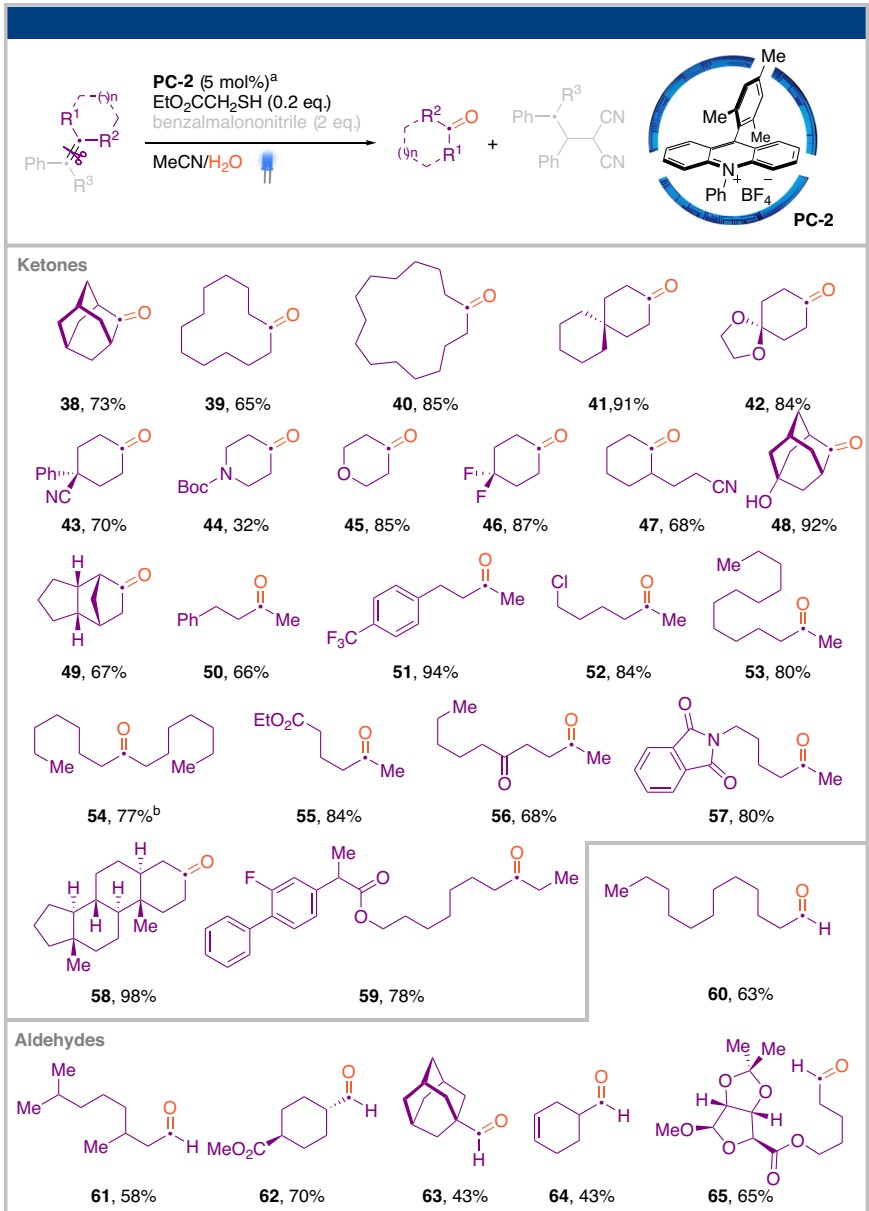

**Fig. 3 | Scope of redox-neutral arylalkene cleavage: the oxidized fragment.** [a]A two-dram glass vial containing a mixture of **PC-2** (0.01 mmol, 5 mol%), arylalkene (0.2 mmol), EtO$_2$CCH$_2$SH (0.04 mmol) and benzalmalonontrile (0.4 mmol) in MeCN/water (1.0 mL/0.1 mL) was placed between two Kessil LED lights (440 nm, 40 W) and vigorously stirred for 24 h. For products **38–59**, R$^3$ = H, for products **60–65**, R$^3$ = Me. [b]The reaction time was extended to 36 hours.

tertiary alkyl chains being well tolerated. Notably, neopentylbenzene **21** was isolated in 73% yield despite the substrate being a highly hindered trisubstituted alkene. The reaction conditions were further challenged with several structurally complex substrates (**22–26**), including those containing chemically labile functional groups. Good to high yields were achieved, demonstrating the robustness of this protocol. Simple, non-aryl alkenes were unable to progress past the hydration step due to the extreme oxidation potential of the resulting alcohol intermediate. In contrast, terminal aliphatic alkenes bearing a phenyl group at the β-position were found to be appropriate substrates when using pentafluorothiophenol as the HAT reagent, resulting in products **35–37**.

The scope of the oxidized carbon was subsequently explored by altering the R$^1$ and R$^2$ groups. The isolation of the oxidized carbonyl fragment necessitated adding benzalmalononitrile (2.0 eq.) as a radical scavenger to suppressing side reactions. Without this additive, the isolated yield of the ketone/aldehyde products was significantly lower

(see Supplementary Information for details). It is worth noting that the isolation of the reduced fragment (Fig. 2) did not necessitate any external radical scavenger. Employing a slightly modified set of reaction conditions, the corresponding carbonyl fragments were efficiently isolated (Fig. 3). Various tri-substituted exocyclic alkenes, including those featuring a spiro or bridged ring scaffold, underwent cleavage to yield cyclic ketone products. Functional groups such as cyano, halogen, ether, alcohol, ketone, carbamate, imide, ester, and others were found to be compatible with the photoredox conditions. Acyclic arylalkenes exhibited reactivity comparable to their cyclic counterparts. Additional aryl substituents did not interfere with the single-electron oxidation (products **50–51**). The corresponding aldehyde products were smoothly generated when either R$^1$ or R$^2$ was hydrogen, despite the aldehyde being a potential quencher of the benzyl radical (products **60–64**)[34]. Structurally complex side chains showed compatibility similar to those in Fig. 2 (products **58–59, 65**).

In Figs. 2 and 3, our primary focus was on isolating only one of the two redox fragments. The isolation of both fragments out of a single reaction is achievable under the conditions described in Fig. 2. The reactions showcased in Fig. 4 represent disproportionative cleavage of arylalkenes, resulting in the formation of one reduced and one oxidized product. Both the corresponding arene and ketone products were isolated in similar yields, which indicates the high chemoselectivity of the HAT process mediated by $EtO_2CCH_2SH$. Kinetic isotope experiments were conducted to gain insights into the reaction mechanism. The results suggest that the initial addition of water to the C=C bond is likely rate-determining, with the HAT from $EtO_2CCH_2SH$ to the benzyl radical being the slowest step ($k_H/k_D = 1.9$; see Supplementary Information for details).

The generation of a reduced product post C=C bond cleavage presents a unique opportunity to access partially deuterated alkylarenes through deuterium-atom-transfer (DAT) processes. In this reaction, one fragment that is reduced undergoes two sequential HAT events, resulting in the formation of a doubly hydrogenated product. When $D_2O$ is employed as a substitute for $H_2O$, the reaction proceeds efficiently, yielding dideuteriomethylated (-$CD_2H$) arenes with good to high yields and notable isotopic enrichment (Fig. 5). While the synthesis of $CD_3$-containing molecules has been extensively studied, there are considerably fewer reports on the development of methods for $CD_2H$ derivatives[35,36]. Our methodology offers a straightforward approach to accessing this intriguing functionality from easily obtainable arylalkenes, making it a valuable tool for medicinal chemistry applications.

To gain further mechanistic insights into the overall transformation, we conducted several control experiments. In addition to forming a reduced fragment, the radical intermediate can also be intercepted by an external radicalophile, such as benzalmalononitrile. The corresponding benzylation product **66** was isolated in 83%, supporting the formation of benzyl radical after C=C cleavage (Fig. 6a). When an anti-Markovnikov water addition intermediate (**14a-OH**) was subjected to the standard reaction conditions, the desired $C_{sp3}$-$C_{sp3}$ cleavage product (**14**) was obtained in a yield similar to that in Fig. 2, supporting the involvement of the alcohol intermediate (Fig. 6b). Alcohols, in addition to water, have proven to be competent nucleophilic additives that can promote alkene cleavage (Fig. 6c). Following the anti-Markovnikov addition of alcohol, the resulting tertiary alkyl ether undergoes a thermochemically-driven $C_{sp3}$-$C_{sp3}$ cleavage in the presence of a potent photooxidant[37]. Interestingly, the process differs from the use of $H_2O$, which proceeds through a PCET to form an oxygen radical intermediate prior to C-C cleavage. The corresponding β-scission involving the radical cation of an alkyl ether is considerably more challenging[38]. Remarkably, a similar efficiency of C=C bond cleavage was observed when water was replaced by an equal volume of MeOH. However, a reduced yield was observed when using isopropanol as nucleophile. In instances where the substrate contains two C=C bonds, the corresponding styrenyl olefin undergoes cleavage with chemospecificity, irrespective of steric factors (Fig. 6d). $^{18}O$-labeled ketone can be prepared in high yield and isotopic enrichment using $^{18}O$-water, as demonstrated in Fig. 6e. The reaction can be conveniently performed on gram scales using a simple RB flask setup, as depicted in Fig. 6f.

## Discussion

In summary, we have developed a redox-neutral approach for the cleavage of C=C bonds present in arylalkenes. This strategy distinguishes itself from conventional oxidative protocols by facilitating the formation of a reduction product following bond deconstruction. We have identified a thiol as a highly chemoselective HAT reagent that effectively couples an anti-Markovnikov addition step with an oxygen radical-mediated $C_{sp3}$-$C_{sp3}$ cleavage step, thus enabling the sequential cleavage of C=C bonds. As a result, arylalkenes undergo disproportionation, yielding one reduction product and one oxidation product. This method for alkene cleavage complements existing techniques in terms of overall changes in oxidation states and accessible functionalities. It demonstrates significant potential for a wide range of applications in synthetic chemistry and the structural modification of functional molecules, thereby contributing to the advancement of the field.

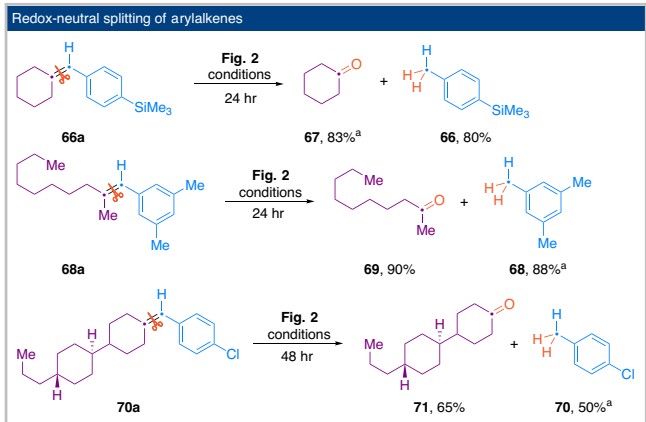

**Fig. 4 | Disproportionative cleavage of arylalkenes.** [a]Due to the low boiling points of these products, yields were determined by GC.

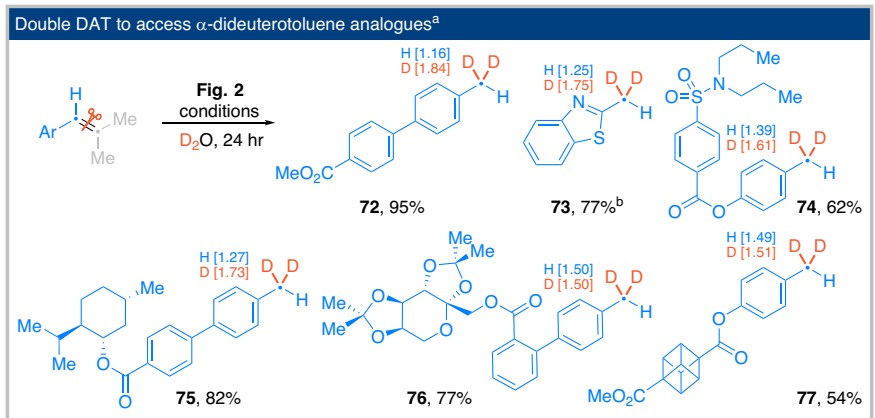

**Fig. 5 | Reductive deuteration experiments.** [a]The reaction conditions employed are identical to those presented in Fig. 2, with the exception of using $D_2O$ as a substitute for $H_2O$. [b]The reaction time was extended to 72 h.

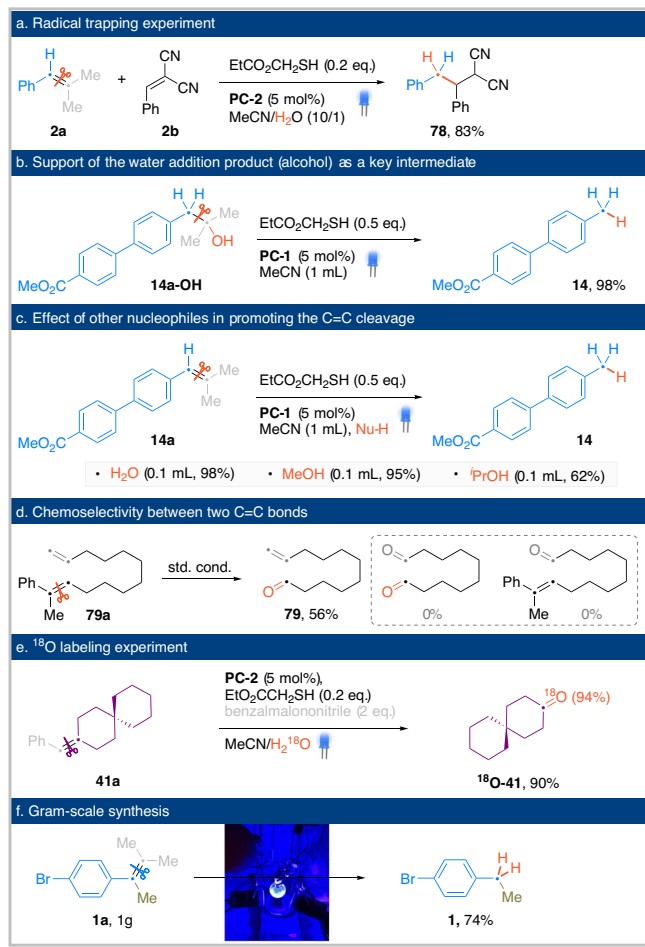

**Fig. 6 | Mechanistic experiments and synthetic utilities. a** Trapping of the resulting benzyl radical using benzalmalononitrile. **b** A control experiment using the corresponding alcohol suggested it being one of the key intermediate. **c** Results of using alcohols as the promotor for the alkene cleavage. **d** Chemoselectivity can be accomplished between a substituted styrene and a simpe alkene. **e** The synthesis of a [18]O-labeled ketone was accomplished. **f** A gram-scale reaction was performed using a round-bottom flask.

## Methods

### General method for isolating the reducing fragment

**PC-1** (0.01 mmol, 5.7 mg) was weighed in an oven-dried 8 mL vial equipped with a magnetic stirring bar. Water (0.1 mL) and MeCN (1.0 mL) were added, followed by $EtO_2CCH_2SH$ (0.1 mmol) and an arylalkene substrate (0.2 mmol). The reaction vessel was degassed, back-filled with argon, and placed in a water bath between two kessil lights (40 W*2). The vial was placed against the wall of the water bath to minimize light reflection (the distance of the Kessil lamps from the vials is about 5 cm, refer to Supplementary Fig. 1 in the Supplementary Information for pictures of the reaction setup). The water bath was maintained at 35 °C to ensure stable temperature control. The progress of the reaction was monitored by TLC. Upon completion, the reaction mixture was concentrated and purified by silica gel flash column chromatography.

### General method for isolating the oxidating fragment

**PC-2** (0.01 mmol, 4.0 mg) and benzalmalononitrile (0.4 mmol) were weighed in an oven-dried 8 mL vial equipped with a magnetic stirring bar. Water (0.1 mL) and MeCN (1.0 mL) were added, followed by $EtO_2CCH_2SH$ (0.04 mmol) and an arylalkene substrate (0.2 mmol). The reaction vessel was degassed, back-filled with argon, and placed in a water bath between two kessil lights (40 W*2). The vial was placed

against the wall of the water bath to minimize light reflection (the distance of the Kessil lamps from the vials is about 5 cm, refer to Supplementary Fig. 1 for pictures of the reaction setup). The water bath was maintained at 35 °C to ensure stable temperature control. The progress of the reaction was monitored by TLC. Upon completion, the reaction mixture was concentrated and purified by silica gel flash column chromatography.

## Data availability

Experimental procedures, characterizations of new compounds are included in the Supplementary Methods. For NMR and HPLC spectra of structurally novel compounds, see Supplementary Figs. All other data are available from the authors upon request.

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

## Acknowledgements

This work was financially supported by Hong Kong RGC (16305523) and by the National Natural Science Foundation of China (21825101).

## Author contributions

Y.H. and J.C. directed the project. K.L., Y.F., and L.S. conducted the experiments. All authors contributed to analyzing the experimental results, as well as to the writing of this manuscript.

## Competing interests

The authors declare no competing interests.
