## [Peer Review File · Nature Communications]

Water Mediated Redox-Neutral Cleavage of Arylalkenes via Photoredox CatalysisReviewers' Comments:

Reviewer #1:

Remarks to the Author:

In the present manuscript, the authors report a radical protocol on the C=C bond cleavage of multi-substituted styrene derivatives. This transformation proceeds through a radical sequence involving the visible-light-promoted anti-Markovnikov hydration, light mediated O-radical formation, and subsequent radical-based beta-scission, achieving the formal carbon-carbon double cleavage. All these "key steps" are well developed (see ref. 25-28 in the manuscript), and the simple combination does not provide significantly enough novelty. The authors claimed the compatibility between O-radical species and the required hydrogen atom transfer reagent would be challenging, however, most S-based HAT reagents used in this work (see Table S2) yielded the desired product. The selective C=C bond cleavage of simple styrene and aliphatic alkenes would be interesting, but this protocol can't do it. This protocol only tolerates alkenes containing multi-substitutions, which are typical valuable chemicals prepared from carbonyl compounds, just like the authors did. Incomprehensibly, the authors prepared multi-substituted alkenes from carbonyl compounds, and then cleaved the C=C bond to back produce carbonyl compounds. The outcome of the transformation is not so convincing. Notwithstanding the consistent results, I did not find the manuscript suitable for Nature Communication. I suggest conveying the submission to more specialised journals in organic chemistry.

Other comments:

1. Why did the authors use 2 equiv of benzalmalononitrile in table 2? What are the yields of the radical trapping products in table 2? What if other radical trapping reagents are used?
2. Several products, such as compound 2 and 12, were determined only by GC due to the low boiling points, however, those compounds were still characterized as colorless oils. How did the author do to conclude it?
3. The title should be more specific.

Reviewer #2:

Remarks to the Author:

Recommendation: Reconsider After Major Revisions.

Comments:

Huang and coworkers have reported a method that cleaves C(sp²) double bonds by intuitively undergoing an anti-Markovnikov hydration step followed by an HAT-assisted beta-fragmentation of a newly formed C(sp³) bond under mild photoredox-neutral conditions, thereby getting both oxidized and reduced products. Below are the strengths and weaknesses of the manuscript

Strengths:

1. The authors introduce a novel system that is easy to set up and it proceeds in the absence of both transition-metals or hazardous reagents.
2. The scope of substrates is decent, and the optimized conditions provide a robust protocol for a decent variety of alkenes.
3. The method plays upon its selectivity to form reduced products of styrenyl compounds and oxidized products of aliphatic compounds in a way that allows for predicted outcomes.
4. The protocol appears somewhat selective (based on very few examples).
5. The mechanistic experiments are well designed to support the most likely pathway of the reaction.
6. The protocol uses water as a reagent and thus has the potential to access isotopic (²H or ¹⁸O) labeling of aliphatic ketones and styrenyl hydrocarbons.

Weaknesses:

7. While the protocol is regioselective, there are no examples of testing unactivated aliphatic alkenes. Nicewicz and many others have illustrated that tri and tetrasubstituted alkenes can be oxidized with an

appropriate PC. I think reporting a few examples of this is highly warranted.

8. There are no indications of a bench-scale (~0.5-1 gram) procedure. From what I can see, all the reactions are conducted at 0.1-0.2 mmol scale. Typically, it is of interest for reactions to have gram scale (at least ~1g) for broader appeal and synthetic use, or at least, be adapted to flow-chemistry at reasonable rates. The authors fail to demonstrate the reactions can be scaled up or adapted to flow. I would recommend the author's show at least 1 scale up of the reaction. If they cannot, this limitation needs to be discussed in the manuscript.

9. The author's use benzylidenemalonitrile as a radical trap for their reactions in their aim to obtain useful aliphatic ketones for the entirety of Table 2. They do not mention, from what I can see, if this is synthetically necessary. If the reaction does not use this stoichiometric trap, based on the scope of Table 1, will you just end with the reduced unwanted styrenyl byproduct? I think a reason should be noted in the manuscript.

10. I think it's prudent to show the limitations of the protocol in the manuscript itself. While this method plays strategically from these limitations to garner desired products, further limitations could affect those who wish to adopt the method for compounds not currently shown in the scope tables. For example, alcohols are well-used and needed moieties for several compounds, intermediates, and drug-candidates. Would having alcohols as part of your target substrate, which would also be prone to HAT, lead to decomposition products? What about other functional groups that could potentially be reduced, such as alkynes and extremely common targets like benzylic ketones (not esters). Scope that includes these functional groups should be added and discussed (whether they failed or not).

11. There is a lack of scope that tries to determine if there is any selectivity for these reactions. There is one example for a competing terminal aliphatic alkene. As for others, there is a cyclic one (52) that provides a yield of 43%. Do you see any cleavage of the internal alkene in this second case. What is the rest of the mass-balance here? It would be of interest to see other sites as well, like internal styrenyl or internal (noncyclic) aliphatic alkenes. This would add to any potential limitations of the protocol or its broad strategic appeal.

12. There is a complete absence of isolation protocols in the SI, which is egregious. Simply putting "column chromatography" is insufficient. Please include the column conditions employed (eluent). Especially considering the volatility of ketones, and doing them at 0.1 mmol scale, the difficulty of isolation at high yields is a concern and can be an issue from a reproducibility aspect. Considering this is a reagent paper of a new method, it is of high necessity that both the ability to implement the reaction and isolate the products are detailed.

13. Unless I missed it, there are no controls listed in the SI. For example, reaction conducted in the dark or with the absence of PC or Additive. While the results may be a given, controls are standard elements of a paper.

SI:

14. Some other notes for the SI.

a. Please include the distance of the Kessil lamps from the vials used.

b. Include a note that a water bath was used for heating. The light from the Kessil lamps can be scattered by the water and glass as it hits the vial. While there is a picture and a note in one of the tables, which is good, including it in the text of the procedure may prove necessary as this could cause potential reproducibility issues.

c. It is common for NMRs to be diluted enough to see the solvent signal. This will ensure that referencing is done correctly. Many spectra do not have the solvent peak picked or visible. Please ensure all spectra are indeed referenced.

d. 14a-OH NMR shows a peak picked and integrated at 1.55 ppm. This is typically the shift for water in chloroform. Are you sure this is the O-H peak of alcohol and not just trace water? Perhaps COSY can

be used to confirm.

Addition Comments:

If possible, an example of isotopic ^{18}O water to obtain valuable ^{18}O -labeled ketones may be a good addition to show synthetic utility.

Table 2, bottom (substrates 49-53) should have a labeled indication as to why that section is divided. Thus, a person can see from the table or its captions why these substrates are special without having to read the paper.

In summary, I recommend to reconsider after major revisions. While there are clear limitations to this protocol, the strategy employed, and its catalytic nature can make it of significant interest and offers a great contribution to the field given the high demand for $\text{C}(\text{sp}^2)$ double bond cleavage reactions. However, I do feel that the weaknesses stated above must be addressed before acceptance.

Reviewer #3:

Remarks to the Author:

In this paper, Huang and co-workers described a novel method to cleave the carbon-carbon bonds in a redox-neutral way. This is an interesting transformation and offers new opportunity for the modification of complex molecules. I recommend the publication of this paper after minor revision:

1: For the reductive deuteration experiments, the deuteration ratio is not satisfactory. Could the author modify the condition and improve it?

2: Under this redox-neutral condition, I am wondering whether free hydroxy and amine groups could be tolerated

Reviewers' Comments to Author (Responses are highlighted in blue):

Reviewer #1:

In the present manuscript, the authors report a radical protocol on the C=C bond cleavage of multi-substituted styrene derivatives. This transformation proceeds through a radical sequence involving the visible-light-promoted anti-Markovnikov hydration, light mediated O-radical formation, and subsequent radical-based beta-scission, achieving the formal carbon-carbon double cleavage. All these “key steps” are well developed (see ref. 25-28 in the manuscript), and the simple combination does not provide significantly enough novelty. The authors claimed the compatibility between O-radical species and the required hydrogen atom transfer reagent would be challenging, however, most S-based HAT reagents used in this work (see Table S2) yielded the desired product.

Our response: Thank you for your constructive comments. Although each elementary step involved in the overall catalytic cycle does have some degree of precedence in the literature. Nonetheless, we contend that the conditions and substrate scope of these steps are significantly different, rendering it challenging to combine them within a single catalytic cycle. The primary difficulty in this regard pertains to managing multiple HAT events, as any disparity would promptly shutdown the entire C-C cleavage process. Consequently, the use of water as an auxiliary to facilitate olefinic bond cleavage has not been established, given multiple transient radical species that are implicated (as depicted in Scheme S5). As indicated in the manuscript, our research has revealed that the structure of the HAT reagent (thiol) is critical in suppressing undesirable back HAT processes. Although several S-based HAT reagents were found to deliver the desired products, the conversion was low for most thiols tested, and most importantly, the substrate scope was very narrow. To further address this issue, we conducted a side-by-side comparison of thiols, as shown below. With the exception of ethyl thioglycolate, other thiols resulted in significantly lower yield, and the reaction stalled at the water addition stage.

The selective C=C bond cleavage of simple styrene and aliphatic alkenes would be interesting, but this protocol can't do it. This protocol only tolerates alkenes containing multi-substitutions, which are typical valuable chemicals prepared from carbonyl compounds, just like the authors did.

Our responses: Thank you for your constructive comments. Following additional optimization, we discovered that pentafluorothiophenol serves as a suitable additive to facilitate the cleavage of a variety of unconjugated terminal alkenes. These findings have been incorporated into the revised manuscript. We extend our gratitude for your valuable comments in this context, specifically, in enhancing the substrate scope.

Incomprehensibly, the authors prepared multi-substituted alkenes from carbonyl compounds, and then cleaved the C=C bond to back produce carbonyl compounds. The outcome of the transformation is not so convincing. Notwithstanding the consistent results, I did not find the manuscript suitable for Nature Communication. I suggest conveying the submission to more specialised journals in organic chemistry.

Our responses: In order to quickly examine the scope of various arylalkenes, certain substrates were prepared from the corresponding ketones. However, thanks to the rich chemistry in olefin synthesis, multiple strategies are available to access various arylalkene substrates, including cross-coupling reactions, olefin metathesis, etc. For instance, the substrate (56a) was synthesized through Ni-catalyzed hydroalkylation of alkynes.

To further demonstrate the synthetic appeal of the ketone synthesis, we prepared the the ¹⁸O-labeled ketone from its natural isotopic isomers. The overall transformation illustrate an intriguing oxygen swapping process. With these additional results, we hope we can convince you that this redox-neutral reaction represents a new solution to perform disproportionative cleavage of Csp²=Csp² bonds, which has remained underexplored.

Other comments:

1. Why did the authors use 2 equiv of benzalmalononitrile in table 2? What are the yields of the radical trapping products in table 2? What if other radical trapping reagents are used?

Our response: Our primary objective in Table 2 was to isolate the oxidized fragment

(carbonyl compounds). We discovered that adding benzalmalononitrile (2.0 eq.) was highly effective in scavenging the transient benzylic radical and suppressing side reactions. Without this additive, the isolated yield was significantly lower. It is worth noting that the isolation of the reduced fragment (Table 1) did not necessitate any external radical scavenger, despite a lower isolated yield of the carbonyl fragment.

Several radical scavengers were examined, and benzalmalononitrile remained as the best radical trap for this reaction. Other electron-deficient olefins were found to be capable of serving as trapping agents as well.

2. Several products, such as compound 2 and 12, were determined only by GC due to the low boiling points, however, those compounds were still characterized as colorless oils. How did the author do to conclude it?

Our response: We apologize for the confusion this caused. The ketone products referred have a low boiling point, and their yield was determined using GC. However, we were able to isolate pure samples of these compounds for NMR purpose by careful evaporation, and they were found to be colorless oil. To ensure consistency, we have made revisions to the SI to align with the GC yield for volatile products.

3. The title should be more specific.

Our response: The title has been revised to “Water Mediated Redox-Neutral Cleavage of Arylalkenes via Photoredox Catalysis”.

Reviewer #2:

Recommendation: Reconsider After Major Revisions.

Comments:

Huang and coworkers have reported a method that cleaves C(sp²) double bonds by intuitively undergoing an anti-Markovnikov hydration step followed by an HAT-assisted beta-fragmentation of a newly formed C(sp³) bond under mild photoredox-neutral conditions, thereby getting both oxidized and reduced products. Below are the strengths and weaknesses of the manuscript

Strengths:

1. The authors introduce a novel system that is easy to set up and it proceeds in the absence of both transition-metals or hazardous reagents.
2. The scope of substrates is decent, and the optimized conditions provide a robust protocol for a decent variety of alkenes.
3. The method plays upon its selectivity to form reduced products of styrenyl compounds and oxidized products of aliphatic compounds in a way that allows for predicted outcomes.
4. The protocol appears somewhat selective (based on very few examples).
5. The mechanistic experiments are well designed to support the most likely pathway of the reaction.
6. The protocol uses water as a reagent and thus has the potential to access isotopic (²H or ¹⁸O) labeling of aliphatic ketones and styrenyl hydrocarbons.

Our response: Thank you for your careful evaluation of the manuscript and for providing an accurate summary of this work. The outcomes of the isotope-labelling experiments can be found below in our responses to your additional comments.

Weaknesses:

7. While the protocol is regioselective, there are no examples of testing unactivated aliphatic alkenes. Nicewicz and many others have illustrated that tri and tetrasubstituted alkenes can be oxidized with an appropriate PC. I think reporting a few examples of this is highly warranted.

Our response: As you correctly pointed out, unactivated aliphatic alkenes can undergo anti-Markovnikov water addition smoothly under our standard reaction conditions. However, the resulting aliphatic alcohols have extreme oxidation potentials that exceed the capability of the Fukuzumi acridinium salts. The attempted to tether the water addition with the Knowles PCET protocol was also unsuccessful (*J. Am. Chem. Soc.* **2019**, *141*, 1457-1462). We believe this limitation necessitated a fundamentally different approach, and we are currently exploring various strategies to address this challenge.

During this investigation, we discovered that non-conjugating, terminal alkenes would undergo efficient C=C cleavage when the HAT reagent was switched to pentafluorothiophenol. In this scenario, a remote aryl group is required to bring the oxidation potential of the corresponding alcohols within a reasonable range. These results, along with subsequent discussions, have been incorporated into the revised manuscript. We extend our appreciation to your insightful comments, which contributed to expanding the substrate scope beyond substituted styrenes.

8. There are no indications of a bench-scale (~0.5-1 gram) procedure. From what I can see, all the reactions are conducted at 0.1-0.2 mmol scale. Typically, it is of interest for reactions to have gram scale (at least ~1g) for broader appeal and synthetic use, or at least, be adapted to flow-chemistry at reasonable rates. The authors fail to demonstrate the reactions can be scaled up or adapted to flow. I would recommend the author's show at least 1 scale up of the reaction. If they cannot, this limitation needs to be discussed in the manuscript.

Our response: Thank you for your constructive comments. A scale-up (1 g substrate) experiment was performed using a RB flask as the reaction vessel. The study demonstrated that the amplification process did not have a detrimental effect on the reaction efficiency, and the yield was consistent with that obtained at smaller scales. The relevant data has been incorporated into the revised manuscript.

9. The author's use benzylidenemalonitrile as a radical trap for their reactions in their aim

to obtain useful aliphatic ketones for the entirety of Table 2. They do not mention, from what I can see, if this is synthetically necessary. If the reaction does not use this stoichiometric trap, based on the scope of Table 1, will you just end with the reduced unwanted styrenyl byproduct? I think a reason should be noted in the manuscript.

Our response: Reviewer 1 also raised the same question. We apologize for the lack of clarity in the original manuscript. The primary objective in Table 2 was to isolate the oxidized fragment. We discovered that adding benzalmalonitrile (2.0 eq.) was highly effective in scavenging the transient benzylic radical and suppressing side reactions. Without this additive, the isolated yield of the ketone/aldehyde products was significantly lower (please see below for side-by-side comparisons). It is worth noting that the isolation of the reduced fragment (Table 1) did not necessitate any external radical scavenger, despite a lower isolated yield of the carbonyl fragment. We have modified the corresponding discussion accordingly.

10. I think it's prudent to show the limitations of the protocol in the manuscript itself. While this method plays strategically from these limitations to garner desired products, further limitations could affect those who wish to adopt the method for compounds not currently shown in the scope tables. For example, alcohols are well-used and needed moieties for several compounds, intermediates, and drug-candidates. Would having alcohols as part of your target substrate, which would also be prone to HAT, lead to decomposition products? What about other functional groups that could potentially be reduced, such as alkynes and extremely common targets like benzylic ketones (not esters). Scope that includes these functional groups should be added and discussed (whether they failed or not).

Our response: Thank you for your insightful suggestions. A range of substrates bearing diverse functional groups were investigated using standard conditions. Although primary and secondary alcohols failed to produce the desired C-C cleavage products, tertiary alcohols proved to be suitable substrates, yielding the desired products in high yields, as illustrated in the figure below. Alkynes were also tolerated, although with moderate yields. Benzylic ketones did not impede the photoredox C-C cleavage. Unfortunately, amines were not compatible due to their low oxidation potentials. In contrast, sulfonamide was tolerated

despite its tendency to form an N-radical under PCET conditions. These data have been incorporated into the revised manuscript. Thank you for helping us expand the substrate scope to include these important functional groups.

11. There is a lack of scope that tries to determine if there is any selectivity for these reactions. There is one example for a competing terminal aliphatic alkene. As for others, there is a cyclic one (52) that provides a yield of 43%. Do you see any cleavage of the internal alkene in this second case. What is the rest of the mass-balance here? It would be of interest to see other sites as well, like internal styrenyl or internal (noncyclic) aliphatic alkenes. This would add to any potential limitations of the protocol or its broad strategic appeal.

Our response: Thank you for your insightful comments and suggestions. For product 52, the reaction was quite complex, and only the target product was successfully isolated. We were unable to identify other side products. We tested substrates containing both a styrenyl and a noncyclic internal aliphatic alkene. The results indicated that good chemoselectivity can be achieved for styrene over aliphatic alkene, although the isolated yield was moderate. We also observed selectivity towards styrenyl over electron-deficient olefins. Unfortunately,

we were unable to achieve selectivity between two electronically biased styrenes, which resulted in messy mixtures. These results and limitations have been incorporated into the revised manuscript.

12. There is a complete absence of isolation protocols in the SI, which is egregious. Simply putting “column chromatography” is insufficient. Please include the column conditions employed (eluent). Especially considering the volatility of ketones, and doing them at 0.1 mmol scale, the difficulty of isolation at high yields is a concern and can be an issue from a reproducibility aspect. Considering this is a reagent paper of a new method, it is of high necessity that both the ability to implement the reaction and isolate the products are detailed.

Our response: We apologize for not providing sufficient isolation details earlier. This information have been amended in the revised SI file. As mentioned earlier, a large-scale (1 gram) reaction was performed to confirm the reliability of this reaction in synthesis.

13. Unless I missed it, there are no controls listed in the SI. For example, reaction conducted in the dark or with the absence of PC or Additive. While the results may be a given, controls are standard elements of a paper.

Our response: These control experiments were performed previously and they are summarized in the SI.

SI:

14. Some other notes for the SI. a. Please include the distance of the Kessil lamps from the vials used.

Our response: The distance of the Kessil lamps from the vials has been specified in the General Procedure of SI.

b. Include a note that a water bath was used for heating. The light from the Kessil lamps can be scattered by the water and glass as it hits the vial. While there is a picture and a note in one of the tables, which is good, including it in the text of the procedure may prove

necessary as this could cause potential reproducibility issues.

Our response: Thank you for the constructive comments. We have modified the text of the procedure in the SI to reflect this point.

c. It is common for NMRs to be diluted enough to see the solvent signal. This will ensure that referencing is done correctly. Many spectra do not have the solvent peak picked or visible. Please ensure all spectra are indeed referenced.

Our response: We have doublechecked all spectra, those too concentrated to show reference peaks have been retaken and the solvent peaks have been annotated.

d. 14a-OH NMR shows a peak picked and integrated at 1.55 ppm. This is typically the shift for water in chloroform. Are you sure this is the O-H peak of alcohol and not just trace water? Perhaps COSY can be used to confirm.

Our response: Your absolutely right! The broad peak at 1.55 ppm was indeed water. Through dilution NMR experiments, it was obvious that the integration ratio of this peak vs other peaks increased with dilution. Findings from our HMBC analysis were also consistent with this conclusion. Thank you very much for identifying this issue. The updated NMR spectrum for **14a-OH** has been revised to rectify this error.

HMBC spectrum of **14a-OH** in CDCl₃

A ¹H NMR of **14a-OH** at different dilutions was compared to identify the peak at 1.55 ppm.

(a) ¹H NMR spectrum of the undiluted **14a-OH**.

(b) ^1H NMR spectrum of the **14a-OH** after first dilution.

(c) ^1H NMR spectrum of the **14a-OH** after second dilution.

Addition Comments:

If possible, an example of isotopic ^{18}O water to obtain valuable ^{18}O -labeled ketones may be a good addition to show synthetic utility.

Our response: Thank you for the insightful suggestion. An ^{18}O -labeled experiment was conducted using **41a** as the substrate. The isotopically enriched ketone ^{18}O -**41** was isolated in 90% yield. This result certainly strengthens the synthetic utility of this method. The data has been added to the revised manuscript.

Table 2, bottom (substrates 49-53) should have a labeled indication as to why that section is divided. Thus, a person can see from the table or its captions why these substrates are special without having to read the paper.

Our response: Annotations have been provided to explain the rationale behind the segregation of products (ketones vs aldehydes).

In summary, I recommend to reconsider after major revisions. While there are clear limitations to this protocol, the strategy employed, and its catalytic nature can make it of significant interest and offers a great contribution to the field given the high demand for $\text{C}(\text{sp}^2)$ double bond cleavage reactions. However, I do feel that the weaknesses stated above must be addressed before acceptance.

Our response: Thank you for the positive comments on the overall significance of this work. We greatly appreciate your invaluable suggestions, which have significantly enhanced the quality and strength of this manuscript.

Reviewer #3:

In this paper, Huang and co-workers described a novel method to cleave the carbon-carbon bonds in a redox-neutral way. This is an interesting transformation and offers new opportunity for the modification of complex molecules. I recommend the publication of this paper after minor revision:

Our response: Thank you for your generous compliments regarding the overall quality of this work. We sincerely appreciate your support and encouragement.

1: For the reductive deuteration experiments, the deuteration ratio is not satisfactory. Could the author modify the condition and improve it?

Our response: Thank you for your insightful suggestions. The probable cause of the observed incomplete deuteration is back HAT processes. To eliminate the possibility of H_2O as a source of H, all reagents and solvents were thoroughly dried. The reactants were mixed in a glovebox. The isotope content for products 62, 63, and 65 was improved. These changes have been incorporated into the revised manuscript's data presentation.

2: Under this redox-neutral condition, I am wondering whether free hydroxy and amine groups could be tolerated

Our response: Thank you very much for your insightful suggestions. A range of substrates bearing diverse functional groups were investigated using standard conditions. Although primary and secondary alcohols failed to produce the desired C-C cleavage products, tertiary alcohols proved to be suitable substrates, yielding the desired products in high yields. Alkynes were also tolerated, although with moderate yields. Benzylic ketones did not impede the photoredox C-C cleavage. Unfortunately, amines were not compatible with due to their low oxidation potentials (*Synlett* **2016**, 27, 714-723; *ACS Catal.* **2020**, 10, 11712-11738). In contrast, sulfonamide was tolerated despite its tendency to form an N-radical under PCET conditions. These data have been incorporated into the revised manuscript. Thank you for helping us expand the substrate scope to include these important functional groups.

Reviewers' Comments:

Reviewer #1:

Remarks to the Author:

My concerns are still there. It is not convincing to be published in Nature Communication. The simple combination of the known protocols does not provide novelty, since the use of other thiols had realized such a combination is not that challenging. That the structure of thiols be critical is not surprising, and the screening of thiols is only a technical problem. Actually, this paper is not the first one to use ethyl thioglycolate to promote HAT in a radical reaction. The authors should not overelaborate the challenging.

I still insist on the previous comment I made. Simple styrene and many aliphatic alkenes can be obtained in large quantities from petrochemicals or biomass. "The selective C=C bond cleavage of simple styrene and aliphatic alkenes would be interesting, but this protocol can't do it." The C=C bond cleavage in this paper using only specific alkenes is not that convincing to be published in such a high-level journal, such as Nature Communications.

The experiment of preparation of ^{18}O -labeled ketone from its natural isotopic isomer is indeed making the reaction more attractive. I would recommend to switch H_2O to ^{18}O -labeled water in Table 2 to directly prepare ^{18}O -labeled ketones.

Reviewer #2:

Remarks to the Author:

The authors have addressed all of the reviewer's comments. The manuscript is now suitable for publication.

Reviewer #3:

Remarks to the Author:

The authors have addressed all the issues, and expanded the substrate scope. Therefore, I recommend the publication of this paper in its current status.

Reviewer #1 (Remarks to the Author):

My concerns are still there. It is not convincing to be published in Nature Communication.

The simple combination of the known protocols does not provide novelty, since the use of other thiols had realized such a combination is not that challenging. That the structure of thiols be critical is not surprising, and the screening of thiols is only a technical problem. Actually, this paper is not the first one to use ethyl thioglycolate to promote HAT in a radical reaction. The authors should not overelaborate the challenging.

I still insist on the previous comment I made. Simple styrene and many aliphatic alkenes can be obtained in large quantities from petrochemicals or biomass. "The selective C=C bond cleavage of simple styrene and aliphatic alkenes would be interesting, but this protocol can't do it." The C=C bond cleavage in this paper using only specific alkenes is not that convincing to be published in such a high-level journal, such as Nature Communications.

The experiment of preparation of ^{18}O -labeled ketone from its natural isotopic isomer is indeed making the reaction more attractive. I would recommend to switch H_2O to ^{18}O -labeled water in Table 2 to directly prepare ^{18}O -labeled ketones.

Our response: Thank you for your constructive comments and for acknowledging the improvements in the revised draft. We would like to further emphasize the impact of this work. Our method represents the first example of redox-neutral cleavage of arylalkenes under mild conditions. Unlike previous protocols that require strong oxidants and yield only two carbonyl products, our reaction is unique due to its disproportionative nature. Although the thiols used in our study were not discovered by us, the reaction's sensitivity to various thiols highlights the challenge associated with multiple hydrogen atom transfers (HAT). The primary goal of our work is not to develop new HAT reagents, but to establish a robust protocol for cleaving arylalkenes under redox-neutral conditions. For substrates currently outside our scope, we are actively exploring different strategies to address them and will report our findings in due course.

Regarding Table 2, we have demonstrated that using ^{18}O -labeled water performs equally well compared to ^{16}O -water (Fig. 4e). There is no mechanistic reason why other substrates would not behave similarly. Therefore, re-running all substrates in Table 2 with ^{18}O -water, in our opinion, is unnecessary. Given the competitive nature of this field, we aim to publish our work as soon as possible.

Reviewer #2 (Remarks to the Author):

The authors have addressed all of the reviewer's comments. The manuscript is now suitable for publication.

Our response: We appreciate your feedback and are delighted to know that you are satisfied with the revisions. We are grateful for your insightful comments, which have greatly enhanced the quality of this work.

Reviewer #3 (Remarks to the Author):

The authors have addressed all the issues, and expanded the substrate scope.

Therefore, I recommend the publication of this paper in its current status.

Our response: We appreciate your feedback and are delighted to know that you are satisfied with the revisions. We are grateful for your insightful comments, which have greatly enhanced the quality of this work.